# PMFFNet: A hybrid network based on feature pyramid for ovarian tumor segmentation

Lang Li[1][☯], Liang He[2][☯], Wenjia Guo[3], Jing Ma[4], Gang Sun[5,6], Hongbing Ma[2][☯]*

1 School of Software, Xinjiang University, Urumqi, Xinjiang, China, 2 Department of Electronic Engineering, Beijing National Research Center for Information Science and Technology, Tsinghua University, Beijing, China, 3 Cancer Institute, Affiliated Cancer Hospital of Xinjiang Medical University, Urumqi, Xinjiang, China, 4 School of Computer Science and Technology, Xinjiang University, Urumqi, Xinjiang, China, 5 Department of Breast and Thyroid Surgery, The Affiliated Cancer Hospital of Xinjiang Medical University, Urumqi, Xinjiang, China, 6 Xinjiang Cancer Center, Key Laboratory of Oncology of Xinjiang Uyghur Autonomous Region, Urumqi, Xinjiang, China

☯ These authors contributed equally to this work.
* hbma@tsinghua.edu.cn

**Data Availability Statement:** We evaluate our algorithm on the public dataset MMOTU and can be accessed from the following source https://doi.org/10.6084/m9.figshare.25058690.

## Abstract

Ovarian cancer is a highly lethal malignancy in the field of oncology. Generally speaking, the segmentation of ovarian medical images is a necessary prerequisite for the diagnosis and treatment planning. Therefore, accurately segmenting ovarian tumors is of utmost importance. In this work, we propose a hybrid network called PMFFNet to improve the segmentation accuracy of ovarian tumors. The PMFFNet utilizes an encoder-decoder architecture. Specifically, the encoder incorporates the ViTAEv2 model to extract inter-layer multi-scale features from the feature pyramid. To address the limitation of fixed window size that hinders sufficient interaction of information, we introduce Varied-Size Window Attention (VSA) to the ViTAEv2 model to capture rich contextual information. Additionally, recognizing the significance of multi-scale features, we introduce the Multi-scale Feature Fusion Block (MFB) module. The MFB module enhances the network's capacity to learn intricate features by capturing both local and multi-scale information, thereby enabling more precise segmentation of ovarian tumors. Finally, in conjunction with our designed decoder, our model achieves outstanding performance on the MMOTU dataset. The results are highly promising, with the model achieving scores of 97.24%, 91.15%, and 87.25% in mACC, mIoU, and mDice metrics, respectively. When compared to several Unet-based and advanced models, our approach demonstrates the best segmentation performance.

## Introduction

Ovarian cancer is a frequent malignant tumor in the female reproductive system, characterized by high lethality and a high incidence of hidden symptoms. With over 300,000 new cases diagnosed annually [1], it ranks as the eighth most common cancer in women. In China, although ovarian cancer ranks third among gynecological malignancies after cervical cancer and endometrial cancer, its mortality rate surpasses the combined mortality of the former two, making

**Funding:** Key Research and Development Project of Xinjiang Uygur Autonomous Region (2022B0319-6). The funders had no role in study design, data collection and analysis, decision to publish, or preparation of the manuscript.

**Competing interests:** The authors have declared that no competing interests exist.

it the leading cause of death among gynecological cancers. This poses a significant threat to women's health. Unfortunately, the incidence of ovarian cancer is increasing year by year. Late diagnosis is a common issue due to a lack of awareness and the absence of effective early screening tests. If ovarian tumors can be detected and diagnosed at an early stage, it will significantly improve the survival rates of ovarian cancer patients.

During the early detection stage of ovarian tumors, it is typically necessary to obtain medical images of the ovarian region as a first step. At present,Commonly used screening techniques include Magnetic Resonance Imaging (MRI), Computed Tomography (CT), two-dimensional (2D) ultrasound scanning and contrast-enhanced ultrasonography (CEUS). Among them, two-dimensional ultrasound scanning is the most widely utilized due to its convenience and minimal impact on the human body. Ultrasound-based imaging methods are commonly used for evaluating ovarian tumors, and tumor segmentation plays an essential role as a preparatory step in the diagnosis and treatment of the disease. Clinical doctors typically perform segmentation on ovarian tumor ultrasound images to identify abnormal areas within the images. Through segmentation, doctors can accurately locate and quantify the position, size, and shape of ovarian tumors. This information helps determine the nature and severity of the lesions, which is crucial for treatment planning, assessing disease progression, and monitoring treatment efficacy. However,ovarian tumor images contain a substantial amount of information, and manually outlining the target area in medical images is time-consuming and labor-intensive for experts. Moreover, due to discrepancies in professional knowledge, clinical experience, physiological fatigue, and subjective differences, consistent outlines of ovarian cancer images among different experts are difficult to achieve. Therefore, the task of automatically segmenting ovarian lesions becomes very challenging. And the purpose of this study is to use artificial intelligence techniques to address the issue of automatic segmentation, aiming to assist clinical doctors in improving diagnostic efficiency.

In contrast to relying solely on the empirical judgment of experts, artificial intelligence technology offers promising solutions in terms of efficiency and accuracy. Early medical image segmentation primarily relied on traditional methods such as edge detection, template matching, region growing, graph cutting, machine learning, and other mathematical techniques. However, With the popularity of deep learning, an increasing number of methods utilizing deep learning have emerged [2–8]. Unet [2] is widely recognized as a foundational framework for medical image segmentation. It employs an encoder-decoder architecture, which integrates features of different scales, and greatly enhances the performance of the segmentation model. Rammah et al. [5] conducted an in-depth exploration of the Unet architecture and optimized the networks of four Unet variants to enhance the performance of brain tumor segmentation. MOHD et al. [6] proposed a novel deep convolutional neural network called DCNNBT for the detection and classification of brain tumors, and achieved impressive results. Kavin et al. [7] implemented an optimal brain tumor classification system by considering the prediction accuracy of AlexNet [9], ResNet 50 [10], and Inception V3 [11]. With the widespread adoption of Transformer [12] in Natural Language Processing (NLP), an increasing number of studies [13–16] have begun using Transformer for medical image segmentation. TransUNet [13] was the first U-shaped network that integrated Transformers into medical image segmentation. By combining Transformers with the U-shaped network, TransUNet outperformed existing models in medical image segmentation at that time. Swin UNETR [15] proposed a self-supervised pretraining segmentation model based on the Swin Transformer [17] architecture. Compared to other advanced models, Swin UNETR achieved the best performance in the BTCV multi-organ segmentation challenge task. The SSFormer [16] network leverages the pyramid encoder PVTv2 [18] to obtain features at different scales, and subsequently employs convolutions to process them. This approach demonstrated notable performance in the colon polyp segmentation task.

When it comes to ovarian cancer tasks, most networks employ Deep Convolutional Neural Networks (DCNNs). For instance, Sakshi et al. [19]utilized a fine-tuned VGG-16 [20] deep learning network to detect the presence of ovarian cysts. Zhang et al. [21] developed an image diagnosis system based on fine-tuned GoogLeNet [22] to classify ovarian cysts in color ultrasound images. Wu et al. [23] explored deep learning methods for ovarian tumor classification based on ultrasound images. Wang et al. [24] employed DCNN and transfer learning techniques to achieve high accuracy in distinguishing between benign, borderline, and malignant ovarian ultrasound images. David et al. [25] proposed a pixel-level cancer segmentation model for ovarian cancer using DMMN [26], and introduced Deep Interactive Learning. However, due to the locality of convolutional operations, convolutional neural networks may not effectively learn global and long-term semantic information interactions. Interestingly, the advantage of the Transformer lies in capturing global information, and the previously mentioned networks that use the Transformer have achieved excellent segmentation performance. In this research, we introduce a network called ViTAEv2 [27], which combines CNN and Transformer, as the encoder part. This allows our model to leverage the Transformer's ability to capture global information and the CNN network's ability to capture local features, thereby enhancing the feature extraction capability. Our final model, PMFFNet, achieves accurate segmentation of ovarian tumors, improving early screening capability and ultimately increasing the survival rate of ovarian cancer patients.

In summary, our work makes the following contributions:

- We introduce the pyramid Transformer ViTAEv2 in our model. Additionally, we address the issue of insufficient information interaction caused by the fixed window size in the ViTAEv2 network by introducing the Varied-Size Window Attention (VSA) design.

- Recognizing the significance of multi-scale features in medical images, we proffer the MFB module. The MFB module aims to learn local and multi-scale features at each scale of the feature pyramid. In this paper, we provide a detailed analysis of this module.

- We have developed a novel feature pyramid-based model called the PMFFNet. We conducted comparative experiments with several Unet-based models and some advanced feature pyramid-based models. Our model outperformed previous research in terms of ovarian tumor segmentation performance.

## Related works

### Transformer based vision backbones

Convolutional neural networks (CNNs) have been the dominant approach in computer vision models for a long time, and many remarkable works have been developed [20, 22]. However, CNNs inherently have limitations. Due to the relatively small convolution kernel size, CNNs primarily rely on local information to understand input images, which can restrict the model's ability to capture comprehensive features. The Transformer-based models have emerged as a solution to overcome this limitation. With the popularity of Transformers in NLP, the Vision Transformer (ViT) [28] introduced the Transformer structure to image classification tasks in computer vision. ViT has demonstrated excellent results and remarkable success across a wide range of computer vision domains when compared to the convolutional neural network architectures. However, ViT does have shortcomings when it comes to downstream tasks. For example, the lack of prior visual information in ViT can lead to slower convergence and lower performance. To address this challenge, researchers have explored methods [17, 29–31] to

integrate prior knowledge from computer vision into Transformer. By combining the strengths of CNNs (such as visual inductive bias like locality and scale-invariance) with Transformer, these hybrid networks have achieved significant advancements in their respective tasks. They serve as effective backbone models for downstream tasks, including semantic segmentation and various other fields, demonstrating impressive performance.

## Feature pyramid networks

Certainly, CNNs [32–34] have been widely used as backbones in computer vision tasks, leveraging features at different scales. Early on, ViT and other similar models operated on a single feature map, without explicitly considering the feature pyramid structure. However, using only the highest-level features for segmentation tasks can lead to a limited receptive field, which may result in the omission of important border-line features. The introduction of the Pyramid Vision Transformer (PVT) [35] marked a significant advancement by incorporating the concept of a feature pyramid into Transformer-based models. This breakthrough inspired the development of various feature pyramid extraction usages [16, 36, 37], which have consistently achieved impressive results across multiple tasks. The combination of feature pyramid extraction and Transformer architectures has thus emerged as a powerful approach, leading to significant advances in the field.

## Multi-scale feature method and dilated convolution

In the context of signal processing, multi-scale refers to the sampling of different granularities in order to complete various tasks effectively. Semantic segmentation, specifically pixel-level classification, benefits greatly from the use of multi-scale features as it requires accurate delineation of object boundaries. There are currently two main categories of multi-scale methods: inter-layer multi-scale structures and intra-layer multi-scale structures. Inter-layer multi-scale structures utilize different feature scales extracted by the encoder and progressively aggregate them in the decoder. Examples of such architectures include U-shaped networks [2–4, 13–15]. On the other hand, intra-layer multi-scale structures typically employ multi-scale modules, such as ASPP [38] and DenseASPP [39]. These models incorporate parallel multi-branch convolutional layers with varying dilation rates to capture rich combinations of receptive fields. In this paper, we adopt the feature pyramid method to obtain inter-layer multi-scale features. Additionally, we leverage the MFB module to obtain intra-layer multi-scale features. By combining both inter-layer and intra-layer multiscale features, we have achieved accurate object localization and precise delineation of fine boundaries. Dilated convolution, introduced by Yu et al. [40], is a technique that incorporates holes or gaps within the standard convolutional map. It is effective in increasing the receptive field of the convolution operation and capturing multi-scale contextual information. The size of the receptive field in dilated convolution represents the network's ability to learn global features. While this is beneficial for capturing global information, it may not be ideal for learning small targets or details. In addition, dilated convolution also has two notable drawbacks. The first drawback is the potential issue of sampling sparsity when using excessive dilation rates. To address this, it is common to use stacked dilated convolutions with increasing dilation rates. By gradually increasing the dilation rate, the network can capture both global and local details. The second drawback is the gridding effect [41] caused by dilated convolution, which can result in the loss of local information. In this paper, to overcome these issues, the MFB module utilizes stacked dilated convolutions with dilation parameters of 1, 2, and 5. This strategy aims to strike a balance between capturing global contextual information and preserving local details by avoiding excessively sparse sampling caused by higher dilation rates.

### Encoder-decoder structure

The introduction of the FCN [42], a fully convolutional neural network, has set a precedent for using convolutional neural networks in image segmentation tasks. This framework laid the foundation for achieving pixel-level segmentation in medical image segmentation. Additionally, it has inspired researchers to explore new ideas and solutions based on encoder-decoder architectures [2, 43, 44]. For instance, UNet and other models have provided solutions for semantic segmentation tasks that require both detailed and semantic information. On the other hand, PSPNet [44] and Deeplab [38] have been proposed to capture contextual information by obtaining local, multi-scale, and even global context. Currently, most architectures obtain various scale features from the encoder and then fuse and up-sample them in the decode to obtain the final segmentation results. However, existing feature pyramid-based encoder-decoder networks [16, 36, 37] up-sample the encoder features to 1/4 of the original size at each pyramid layer. For instance, consider an input image size of 384×384. In order to match 1/4 of the original size up-sampling to 96×96 is required. However, the deepest feature size is 12×12. If the feature map is up-sampled from 12×12 to 96×96, it would require an 8-fold up-sampling. This high up-sampling factor can potentially result in a loss of detailed information in the process. To address this, we propose placing the up-sampling operation in the decoder, up-sampling the deep features to the scale of the previous layer, and ultimately obtaining the segmentation map. This ensures that the details are preserved.

## Methods

In our model, the encoder-decoder architecture is employed, as illustrated in Fig 1. The encoder utilizes the ViTAEv2+VSA architecture, which we abbreviate as V2S in this context, to extract essential segmentation features pyramid. To further refine the features and increase their receptive fields, the MFB (Multi-scale Feature Fusion Block) module is applied in each layer of the encoder. This module enhances the features by incorporating multi-scale contextual information, improving the model's ability to understand both local and global context. Subsequently, in the decoder of the network, the MFB features are up-sampled, gradually increasing the resolution. Finally, the prediction head utilizes the up-sampled features to generate the final segmentation results.

### Encoder

The encoder employs the V2S architecture to generate a feature pyramid. Each scale of features is fed into the corresponding MFB module.

In this paper, we adopt the ViTAEv2 model [27] as the encoder to capture inter-level feature pyramid. Inspired by Swin Transformer, another inductive bias introduced in ViTAEv2, namely the local window attention. Although attention within windows has been widely explored in vision transformers to balance performance and computational complexity, current models adopt hand-crafted fixed-size window designs, as shown on the left side of Fig 2. This restricts their ability to model long-range dependencies and adapt to objects of different sizes. As shown on the right side of Fig 2, if the windows can be relaxed to different-sized rectangular windows, with window size and position learned directly from the data, the model can capture rich context from different windows and learn more powerful object feature representations. While it requires some additional parameters and FLOPs, the extra computational cost is negligible.

Therefore, addressing the limitations of ViTAEv2 which use fixed-size windows of local window attention, we have incorporated the Varied-Size Window Attention (VSA) [45] into the ViTAEv2 architecture. VSA introduces a window regression module that predicts the size

**Fig 1. Network structure of PMFFNet.**

and location of target windows, allowing for adaptive window configurations that can better handle objects of different sizes. This combination enables the model to effectively model long-term dependencies, capture rich context from different windows, and facilitate information exchange between overlapping windows.

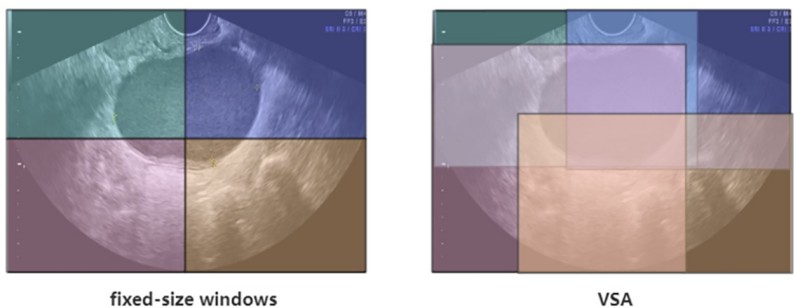

fixed-size windows          VSA

**Fig 2. The comparison between fixed-size windows of local window attention and Varied-Size Window Attention.**

As a result, the combined model is well-suited as the encoder part of the PMFFNet, allowing for the extraction of a feature pyramid. V2S is pretrained on ImageNet, with the ViTAEv2-S variant being used. The V2S architecture consists of four stages, where four RCs are employed to down-sample the features by 4×, 2×, 2×, and 2×, respectively. On the other hand, NC is utilized to model positional and long-term dependencies among tokens. Since the NC module provides richer and more comprehensive information about long dependencies, we obtain the pyramid feature at the NC stage. These pyramid features are then fed into the MFB module for further processing.

## MFB

Our MFB module consists of two main parts: the parallel feature extraction part and the feature fusion part. The parallel feature extraction part is further divided into two subparts: the expanded receptive field module (ERF) and the local emphasis module (LE). The ERF component aims to capture rich global boundary information by expanding the receptive field of the features. This allows the module to gather contextual information from a broader region, enabling better understanding of global structures and boundaries in the image. On the other hand, the LE focuses on extracting enhanced local detail features. It aims to emphasize and preserve fine-grained information in the features. These two subparts work together to capture both global and local information, enabling the MFB module to fuse complementary features that are important for accurate segmentation. The feature fusion part consists of two RB blocks to obtain the final output of the MFB module.

**ERF component.** Given the challenging nature of ovarian cancer medical images, where the boundaries are often unclear, capturing multi-scale features becomes crucial for accurate segmentation. A larger receptive field is essential in this context. The receptive field refers to the effective region of an individual neuron, indicating the range of feature maps it can utilize. A larger receptive field encompasses a broader context and is more likely to incorporate deep features from global and semantic layers. On the other hand, a smaller receptive field tends to focus on more localized and detailed features. By considering the size of the receptive field, we can roughly assess the level of abstraction captured by each layer.

In order to significantly increase the receptive field, we employ dilated convolutions. However, when cascading multiple dilated convolutions with the same dilation parameters, it can lead to a problem known as the gridding effect [41]. For instance, if we use three consecutive dilated convolutions with a dilation parameter of 2 (i.e., dilation parameters of 2, 2, 2), one pixel on the output of the final dilated convolution would not have utilized consecutively adjacent pixel values from the input feature map. This means that a pixel on the final output only incorporates a portion of the available pixel values within its range, resulting in the loss of some detailed information. Consequently, using the same dilation parameter in cascaded dilated convolutions is not advisable. To address this issue, the paper [41] presents a concept known as Hybrid Dilated Convolution (HDC) with specific design guidelines. These guidelines offer suggestions into choosing the dilation parameters for cascaded dilated convolutions:

1. When stacking Z convolution layers with dilation parameters $[d_1, d_2, \ldots, d_i, \ldots, d_z]$ and kernel size K, our objective is to ensure that the stacked dilated convolutions completely cover the underlying feature layer without any holes or missing edges. It defines the "maximum distance between two non-zero values" as follows.

$$M_i = max(M_{i+1} - 2d_i, 2d_i - M_{i+1}, d_i) \tag{1}$$

Where $M_i$ is the maximum distance between two non-zero elements in the ith layer; $d_i$ is

the dilation parameters of the ith layer. For the last layer, it has a maximum distance of $M_z = d_z$. That is, the maximum distance is the dilation parameters of the layer. The purpose of this design is to make the maximum distance between two non-zero elements of the second layer less than or equal to the size of the convolution kernel.

2. Set the dilation parameters to a zigzag shape such as 1, 2, 3, 1, 2, 3.

3. The common divisor of the dilation parameters cannot be greater than 1.

We have dealt with the above suggestions as follows:

For suggestion 1: In this paper, we use kernel size of 3 for the dilated convolutions. When $K = 3$, the dilation parameters $d = 1, 2, 5$. We can apply the formula for Eq (1) as shown in Eq (2). Since $2 \leq 3$, the design requirements are rational.

$$M_2 = max(M_3 - 2d_2, M_3 - 2(M_3 - d_2), d_2) = max(1, -1, 2) \tag{2}$$

In Fig 3, the output represents the result of three stacked dilated convolutions. Each convolution has different dilation parameters: 1, 2, 5, respectively. The convolution kernel size for all layers is $K = 3$. The numbers displayed on each pixel indicate the number of times it was utilized, obtained by accumulating the input image pixel values. Fig 3(a) displays the output of the first layer, which utilizes an ordinary convolution with a dilation of 1. In this case, all pixels within the convolution kernel size window of the input image are used for the computation. Fig 3(b) illustrates the output of the second layer. With a dilation of 2, each pixel in the output corresponds to a 7×7 area on the input image. Although the number of times each pixel is utilized may vary within this area, all pixels are incorporated in the computation. Fig 3(c) depicts the output of the third layer. With a dilation of 5, each pixel in the output corresponds to a 17×17 area on the input image. Similarly, the number of times each pixel is utilized within this area may differ, but all contribute to the overall computation. These results demonstrate that after three stacked dilated convolutions with dilation parameters of 1, 2, 5, respectively, the receptive field of the final layer encompasses all the pixel information from the input. Every pixel's input is used to produce the output. By using the dilation parameters configuration above, the network is able to capture a wider context and significantly expand the receptive field, ensuring comprehensive utilization of pixel values from the input image.

Another question, why does our dilation parameter start from 1, because we hope that each pixel in the high-level feature map can use all the pixels in the receptive field of the bottom feature map. Therefore $M_1$ should be equal to 1. It means that the non-zero elements are adjacent (without gaps). In this paper, the ERF module consists of two cascaded DB blocks. Each DB block is further composed of three cascaded CB blocks. CB is composed of group normalization [46], SiLU activation functions [47] and residual connections [10]. The convolution kernel size for each layer is set to 3×3. The key difference between ERF and LE lies in the



**Fig 3. The output maps of cascaded dilated convolution.**

utilization of the dilation parameter in the CB block. By leveraging different dilation parameters in the CB blocks within the DB, the ERF architecture is able to capture multi-scale information and obtain diverse receptive fields. It is worth noting that the dilation parameters used in this paper are all 1, 2, 5. Consequently, the design of the ERF module, which stacks two DB blocks and each DB block with dilation parameters of 1,2,5, adheres to all the recommended guidelines and can be considered reasonable.

**Local emphasis component.** In this specific module of the architecture, there are two cascaded RB blocks, each composed of two cascaded CB blocks. As the Transformer model becomes deeper, there is a tendency for global features to mix and converge, leading to attention dispersion [48]. Attention dispersion refers to a phenomenon where the Transformer model struggles to accurately predict detailed information in intensive prediction tasks such as semantic segmentation. To address this issue, SSFormer [16] demonstrated a solution that utilizes the local receptive field of the convolutional kernel to increase the macro weights of the patches around the query patch. This refocuses attention on neighboring features, effectively reducing attention dispersion. By emphasizing critical local features and mitigating cluttered noises, this approach helps the model capture and incorporate more detailed information, enhancing its ability to accurately predict in tasks like semantic segmentation.

**Visual analysis.** In the feature map visualization analysis of important modules in PMFFNet, the output features of V2S, LE, ERF, and MFB are analyzed. In the Fig 4, the top is the first scale, and the bottom is the second scale. Due to the increasing abstraction of advanced features, the observation mainly emphasizes first and second scales. Fig 4 visually demonstrates the effectiveness of our model by showing the feature maps. The LE module in PMFFNet is capable of further enhancing the output feature map of the V2S module, effectively making the lesion features more prominent. By incorporating cascaded dilated convolutions, the ERF module expands the receptive field and captures global information more comprehensively. The MFB module then fuses the global information from ERF with the local information from LE. This fusion process results in more accurate and clear lesion boundaries. The combination of ERF and LE modules in PMFFNet allows for the effective integration of global and local information, leading to enhanced lesion detection and segmentation performance.

In the heat map visualization analysis of important modules in PMFFNet, the focus is on observing whether the model effectively captures the characteristics of lesions. The heat maps of LE, ERF and MFB at the second and third scales are analyzed, representing shallow and

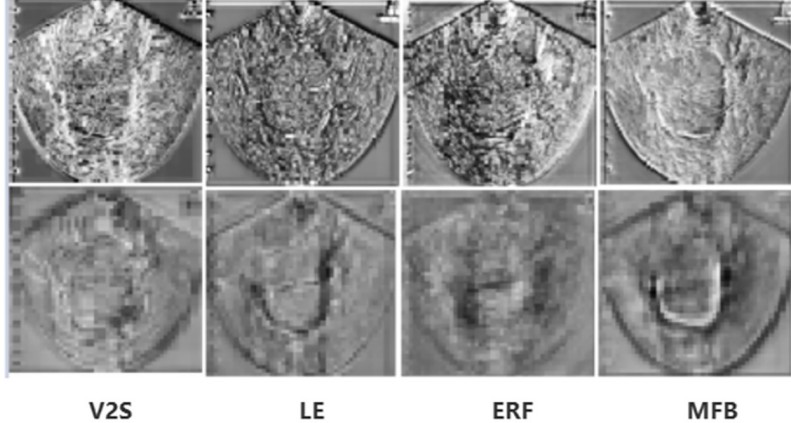

**Fig 4. Feature maps of V2S, LE, ERF, and MFB.**

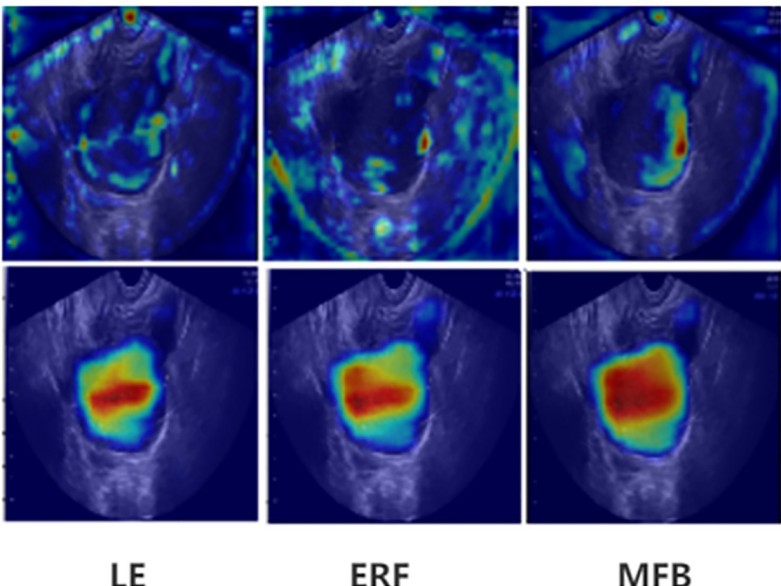

**Fig 5. Heat maps of LE, ERF, MFB.**

deep features, respectively. Fig 5 illustrates these heatmaps. In the first row, representing the shallow features, the LE module shows attention on both the global boundary and the lesion area, with a stronger emphasis on local features within the lesion area. The ERF module also pays attention to the boundary and lesion area, but with a larger receptive field, it focuses more on global information compared to the LE module. After fusion in the MFB module, the attention is focused on the correct and more challenging boundary information, while giving less attention to non-lesional features. This indicates that the MFB module can effectively concentrate on the relevant lesion features while minimizing attention on irrelevant regions. In the second row, representing the deep features, the LE module exhibits attention patterns that are similar to the ground truth, with a particular focus on the deeper red attention area in the middle. The ERF module spreads the attention outward through a larger receptive field, resulting in a wider range of attention compared to LE. Additionally, the boundary attention is smoother and more consistent with the ground truth. After fusion in the MFB module, the focus on the correct lesion features in the middle becomes wider and deeper, and the attention to the boundary becomes wider and finer. This demonstrates that the MFB module has the ability to learn and integrate local features and global boundary information effectively, ultimately leading to a more accurate segmentation map.

## Decoder

In the current network architecture, which is based on feature pyramid (e.g., SSFormer, FCBFormer, SegFormer), all inter-layer scales are up-sampled to 1/4 of their original size. However, this operation can result in the loss of certain information. To address this issue, an up-sampling operation is employed in the decoder, followed by continuous fusion with prior inter-layer scale features to obtain a finer segmentation map. The decoder first utilizes up-sampling to increase the output feature map size of the last MFB module to match the previous scale, which is 1/16 of the original size. This up-sampled feature map is then concatenated with the corresponding inter-layer scale features and passed through two RB blocks. The first RB

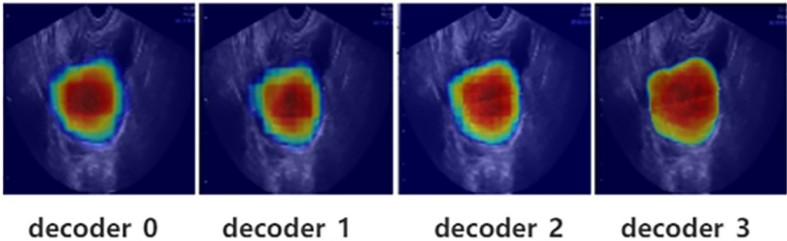

**Fig 6. Heat maps of decoder.**

block has an input channel of 128 and output channel of 64, while the latter has an output channel of 64. The final operation of the decoder is up-sampled to the size of the original input image. Throughout this process, the output channel remains at 64.

Finally, the output of the decoder through two RBs and a 1x1 convolution as prediction head to obtain the final segmentation map for ovarian cancer. The entire process is illustrated in Fig 1, with the pink arrow representing the up-sampling operation. Here we also conducted heat map analysis on the decoder. As shown in Fig 6, decoder_0-2 represent the heat maps after two RBs for three up-sampled feature, and decoder_3 represents the heat map of the final prediction head. Decoder_0 combines the features of the deepest MFB module and the third level inter-scale features. Decoder_1 concatenate the second level inter-scale features with the features from decoder_0. From the heat maps of decoder_0 and decoder_1, we can observe that although the heatmaps focus on the main segmentation features, the boundary information is missing and not smooth enough. This is why we need to fuse information from different inter-scale levels. Decoder_2 combines the first level inter-scale features with the features from decoder_1, and at this point, the contour boundary information becomes clearer, with the red attention extending outward. Decoder_3, which is the heat map of the prediction head, shows a more complete attention to the boundary information, with a uniform distribution of red regions and minimal yellow regions. This indicates that our model can effectively focus on the lesion area and generate accurate segmentation results.

## Loss function

In this medical segmentation task, a composite loss function is utilized, which comprises two components: binary cross-entropy (BCE) loss and Dice loss. The Dice loss focuses on the global perspective by measuring the similarity between the predicted segmentation mask and the ground truth. It is commonly used in segmentation tasks to evaluate the overlap or intersection between the predicted and ground truth regions. On the other hand, the BCE loss considers the gap between the predicted pixel values and the corresponding ground truth labels on a microscopic level.

$$BCELoss(A, B) = -(A \log (B) + (1 - A) \log (1 - B)) \tag{3}$$

$$DiceLoss(A, B) = 1 - \frac{2|A \cap B| + smooth}{|A| + |B| + smooth} \tag{4}$$

$$TotalLoss = BCELoss + DiceLoss \tag{5}$$

BCE loss defined as Eq (3), where A and B represent the underlying ground truth and the

predicted respectively. Dice loss shown in Eq (4), where $|A \cap B|$ represents the intersection of sets A and B, and $|A|$, $|B|$ represent the size of its set elements respectively. For segmentation tasks, $|A|$ and $|B|$ represent the ground true and predict masks of the segmentation, respectively. The total loss function, as depicted in Eq (5). It is added by BCE loss and Dice loss. In this paper, we combine these two loss functions to provide an overall measure of the model's performance.

## Results and analysis

To assess the learning ability and generalization ability of our model, we conducted experiments using the MMOTU [49] public dataset. Specifically, we utilized the OTU_2D subset within this dataset to evaluate our model's learning ability on the same dataset. Additionally, we employed the OTU_CEUS ovarian cancer dataset, which differs from OTU_2D, to evaluate our model's generalization ability on a different datasets. In evaluation, we compared our model against various segmentation methods. Firstly, we compared it with several advanced Unet-based segmentation methods including U-Net, Unet++, ResUNet++, and TransUnet. Additionally, we com-pared our model to more recent advanced segmentation models such as M$^2$SNet [50], FCBFormer, SegFormer, and SSFormer. The results of our experiments demonstrated that our model achieved the best segmentation performance on the MMOTU dataset compared to the aforementioned models. Moreover, our model exhibited superior generalization ability when tested on the OTU_CEUS dataset, which validates its effectiveness in handling different datasets.

In the ablation experiment, we further substantiated the effectiveness of our pro-posed model by splitting our model training on OTU_2D, and we also added the generalization experiment on the OTU_CEUS dataset.

### Experimental setup

**Dataset.** The dataset we employed in our study is sourced from the MMOTU image dataset, which comprises ovarian ultrasound images collected from Beijing Shijitan Hospital, Capital Medical University. Additionally, according to the privacy policy, the MMOTU dataset does not contain any personal information. This dataset includes a total of 1639 ultrasound images from 294 patients. Specifically, the MMOTU dataset consists of two subsets: OTU_2D and OTU_CEUS. The OTU_2D subset comprises 1469 2D ultrasound images, including 216 Color Doppler flow images (CDFI) and 1253 conventional 2D ultrasound images. On the other hand, the OTU_CEUS subset contains 170 images extracted from CEUS sequences.

To facilitate our evaluation process, initially, we split the OTU_2D dataset into a training set and a test set. We allocated 80% of the OTU_2D training set for model training and reserved the remaining 20% for validation purposes. The OTU_2D test set was solely utilized to evaluate the learning ability of our model on the same dataset. The OTU_2D dataset encompasses eight typical ovarian tumor categories, namely Chocolate cyst, Serous cystadenoma, Teratoma, Theca cell tumor, Simple cyst, Normal ovary, Mucinous cystadenoma, High grade serous. However, since our task focuses on segmenting the background and lesion regions, we excluded the Normal ovary samples from the OTU_2D dataset to ensure the reasonable experiment. Conversely, the OTU_CEUS dataset does not include any normal ovary samples and thus required no such processing.

Table 1 summarizes the eventual data setup, providing an overview of the experiment dataset distribution. We included a larger number of test data images to prioritize the quality and comprehensiveness of our evaluation.

**Table 1. The experimental configuration of the MMOTU dataset.**

| Data Type | Training Set | Validation Set | Testing Set | Generalization Set |
|---|---|---|---|---|
| OTU_2D | 656 | 164 | 382 | —————— |
| OTU_CEUS | —————— | —————— | ————— | 170 |

**Evaluation matrix.** In medical segmentation, several evaluation metrics are commonly used to assess the performance and accuracy of segmentation models. The evaluation metrics are as follows.

- The accuracy rate (ACC) indicates the percentage of the number of correctly predicted samples in all samples. That is, the number of correctly classified pixels in the total number of pixels. The formula is defined as Eq (6).

$$Acc = \frac{TP + TN}{TP + TN + FP + TN} \tag{6}$$

- Intersection of Union (IoU) has been used as a standard metric in semantic segmentation. IoU is the ratio of the intersection and union of the model's prediction results for a certain class and the target value. The formula is defined as Eq (7).

$$IoU = \frac{TP}{TP + FP + FN} \tag{7}$$

- The Dice is derived from the binary classification, which essentially measures the overlap of two samples. Where $|X \cap Y|$ represents the intersection of sets $X$ and $Y$, and $|X|$, $|Y|$ represent the ground true and predict masks of the segmentation, respectively. When we implement the code, we often add a smooth. In this work, the default smooth is 1 to prevent the denominator from being 0. Therefore, the formula shown in Eq (8).

$$Dice(X, Y) = \frac{2|X \cap Y| + smooth}{|X| + |Y| + smooth} \tag{8}$$

- Precision (Pr) refers to the proportion of samples predicted by the model as positive examples that are truly positive examples. The formula is defined as Eq (9).

$$Pr = \frac{TP}{TP + FP} \tag{9}$$

- Recall (Re) indicates how many percentages of all samples with positive labels are predicted. The formula is defined as Eq (10).

$$Re = \frac{TP}{TP + FN} \tag{10}$$

Where $TP$, $TN$, $FP$, and $FN$ are pixel-level metrics representing true positive, true negative, false positive, and false negative values in the confusion matrix, respectively. In this work, we use five evaluation metrics, mAcc, mDice, mIoU, mPr and mRe to evaluate segmentation

performance of the proposed model on ovarian tumors dataset. where m indicates an average of the metric value over the test set.

**Implementation details.** Our proposed network is implemented using the Pytorch deep learning framework. The MMOTU dataset contains images with varying scales. To ensure consistency, we resize the input images to a resolution of 384×384. Due to the limited amount of data in the MMOTU dataset, we apply several data augmentation methods such as horizontal flip and vertical flip. For training the network, we employ the binary cross-entropy (BCE) and Dice loss functions as the optimization objectives. We train and evaluate the performance of our network on the OTU_2D dataset, and subsequently assess its generalization capabilities on the OTU_CEUS dataset. The model is trained for 100 epochs using a batch size of 8 on an NVIDIA GeForce RTX 3090 GPU. We employ the AdamW optimizer [51] with an initial learning rate of 1e-4. Specifically, if the metric mDice on the validation set no longer shows improvement for 10 consecutive epochs, the learning rate is reduced by half. The best performing model on the final validation dataset is selected as the final model. To ensure a fair comparison, we set the parameters of all experiments to the same configuration.

## Comparative experiment

To comprehensively assess the performance of our model, we conduct a comparative analysis in two stages.

1. Comparative analysis with U-Net and its variants
   We choose the original U-Net model and its popular variants, including Unet++, ResUnet++, and TransUnet to compare their performance with our proposed PMFFNet architecture. This comparison highlights the advancements of our model with the U-Net series of architectures.

2. Comparative analysis with recently advanced segmentation models
   We further compare the performance of our model with recently advanced segmentation models such as $M^2$SNet, FCBFormer, SegFormer, and SSFormer. This analysis allows for a comprehensive evaluation of our model against the latest advancements in medical image segmentation.

During the evaluation of model performance, we conduct two sets of experiments. Firstly, we assess the learning ability of our model using the OTU_2D dataset. This evaluation focuses on validating the model's capability to accurately segment images within the specific dataset it was trained on. Subsequently, we test the generalization ability of our model on the OTU_CEUS dataset, which contains different characteristics compared to the training dataset. This assessment aims to measure how well our model can adapt and perform on diverse datasets.

**Comparative analysis with U-Net and its variants.** Since there are numerous medical semantic segmentation networks based on the U-Net architecture, the overall performance is often excellent. Therefore, we conduct-ed a comparative analysis to evaluate our model against other U-Net-based architectures. The results in Table 2 are obtained from the OTU_2D test set, and it is evident that our model achieves superior performance compared to the U-Net architecture. Specifically, in terms of the mIoU metric, our model surpasses U-Net, Unet++, ResUNet++, and TransUNet by 11.37%, 7.53%, 6.12%, and 2.87%, respectively. Similarly, in the mDice metric, our model outperforms these architectures by 12.79%, 8.18%, 7.23%, and 3.15%, respectively. These performance improvements strongly validate the effectiveness of our proposed model. Furthermore, it is worth noting that our model exhibits the fewest

**Table 2. Results of comparing with U-Net and its variants on OTU_2D.**

| Model | Parameters | mAcc(%) | mIoU(%) | mDice(%) | mPr(%) | mRe(%) |
|---|---|---|---|---|---|---|
| UNet [2] | 31 M | 93.24 | 79.78 | 74.46 | 95.06 | 96.77 |
| Unet++ [3] | 36 M | 94.77 | 83.62 | 79.07 | 96.38 | 97.32 |
| ResUNet++ [4] | **14 M** | 95.32 | 85.03 | 80.02 | 97.03 | 97.34 |
| TransUnet [14] | 105 M | 96.40 | 88.28 | 84.10 | 97.74 | 97.91 |
| PMFFNet | 23 M | **97.24** | **91.15** | **87.25** | **98.18** | **98.55** |

parameters, except for ResUNet++. Although ResUnet++ has fewer parameters than our model, our performance surpasses it by a large margin. In comparison to TransUNet, the number of parameters in our model is only 22% of its total. Additionally, in our experiments, our model has a FLOPs (floating point operations) of 99G and an inference time of 28 seconds on the test set, while TransUnet has a FLOPs of 178G and an inference time of 45 seconds on the same test set. Overall, these results demonstrate that our model not only achieves superior performance compared to U-Net and its variants but also showcases a notable advantage in terms of parameter efficiency.

We mainly do generalization experiments in order to be able to apply to better generalization performance rather than training better. The experimental results presented in Table 3 are obtained from the OTU_CEUS dataset, while the training was conducted on the OTU_2D dataset. The purpose of these experiments was to assess the generalization ability of our models across different datasets, enabling a more comprehensive evaluation of their performance. As shown in Table 3, our model outper-forms other models in four metrics. Although Unet++ achieves a higher mRe score than our model, its mPr is the lowest among all models and falls significantly behind. Interestingly, our model surpasses the Unet-based model by a large margin in terms of mIOU and mDice metrics. Specifically, our model records a 14.87%, 20.82%,7.56%, and 5.77% increase in mIOU metric compared to U-Net, Unet++, ResUNet++, and TransUNet, respectively. Additionally, our model demonstrates a 17.62%, 19.25%, 11.11%, and 5.79% increase in mDice metric when compared to the aforementioned models. These results validate that our model exhibits superior generalization ability across different datasets compared to the Unet-based architecture.

To conduct a comprehensive analysis of each model's performance, we compared the segmentation results. Fig 7 illustrates the testing set results of training on the OTU_2D dataset in the upper part, while the lower part displays the OTU_CEUS testing results of OTU_2D training.

As shown in Fig 7, observing the images and ground truth of the OTU_2D and OTU_CEUS, it becomes evident that there is a substantial difference between them, making it challenging to accurately segment the images. However, our model rises to this challenge and demonstrates a close match to the ground truth, while other models exhibit a notable number of misjudgments.

**Table 3. Results of comparing with U-Net and its variants on OTU_CEUS.**

| Model | Parameters | mAcc(%) | mIoU(%) | mDice(%) | mPr(%) | mRe(%) |
|---|---|---|---|---|---|---|
| UNet [2] | 31 M | 82.89 | 62.16 | 60.40 | 85.82 | 92.47 |
| Unet++ [3] | 36 M | 75.83 | 56.21 | 58.77 | 72.55 | **96.68** |
| ResUNet++ [4] | **14 M** | 88.66 | 69.47 | 66.91 | 94.48 | 91.75 |
| TransUnet [14] | 105 M | 88.58 | 71.26 | 72.23 | 91.61 | 94.12 |
| PMFFNet | 23 M | **91.65** | **77.03** | **78.02** | **94.86** | 94.82 |

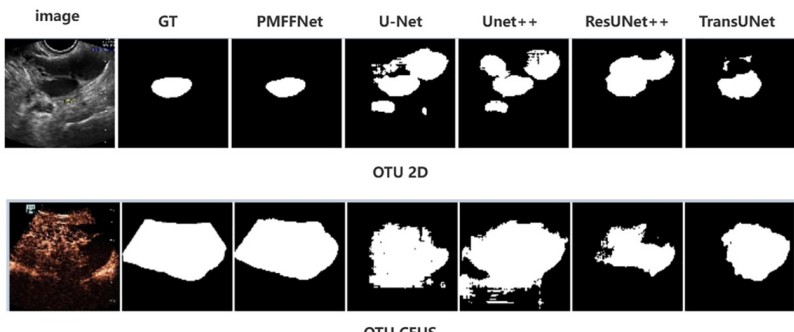

**Fig 7. The segmentation results of U-Net, its variants and our model.**

This observation underscores the powerful learning ability and generalization capability of our model.

**Comparative analysis with recently advanced segmentation models.** Due to the limited research available on ovarian cancer segmentation tasks, so we chose two models from polyp cancer segmentation. Specifically, we selected two models, namely SSFormer and FCBFormer, which have demonstrated exceptional performance in polyp cancer segmentation. Additionally, we included the SegFormer model because the author of the MMOTU using SegFormer achieved outstanding results when evaluating the MMOTU dataset. Lastly, we incorporated the latest model called M²SNet, which proposes a general multi-scale subtraction network. This model effectively captures extreme multi-scale information by utilizing a multi-scale subtraction module, enabling accurate medical image segmentation.

According to Table 4, we can observe that our model continues to outperform advanced models in four metrics, albeit by a relatively small margin. Specifically, when compared to M²SNet, FCBFormer, SegFormer, and SSFormer, our model achieves a higher mIoU metric by 1.86%, 0.89%, 1.29%, and 1.07%, respectively. Furthermore, our model demonstrates a higher mDice metric, surpassing M²SNet, FCBFormer, SegFormer, and SSFormer by 1.92%, 0.64%, 0.45%, and 0.9%, respectively. Despite the relatively small performance differences, it is noteworthy that our model possesses the fewest parameters among all the models evaluated. In fact, it only comprises 28% of the parameter count of SegFormer.

It is important to highlight that the experimental results presented in this experiment are obtained from the OTU_CEUS dataset, while the training is conducted on the OTU_2D dataset. Table 5 displays the performance of our model compared to current advanced models across different datasets. It is evident that our model exhibits exceptional generalization ability, outperforming other models by a significant improvement in four metrics. Specifically, when compared to M²SNet, FCBFormer, SegFormer, and SSFormer, our model achieves a notably higher mIoU metric with improvements of 4.85%, 3.18%, 1.63%, and 1.49% respectively.

**Table 4. Results of comparing with current advanced networks on OTU_2D.**

| Model | Parameters | mAcc(%) | mIoU(%) | mDice(%) | mPr(%) | mRe(%) |
|---|---|---|---|---|---|---|
| M²SNet [44] | 29 M | 96.77 | 89.29 | 85.33 | **98.21** | 97.90 |
| FCBFormer [29] | 52 M | 97.13 | 90.26 | 86.61 | 98.20 | 98.34 |
| SegFormer [30] | 84 M | 96.94 | 89.86 | 86.80 | 98.01 | 98.29 |
| SSFormer [17] | 66 M | 97.11 | 90.08 | 86.35 | 98.17 | 98.35 |
| PMFFNet | **23 M** | **97.24** | **91.15** | **87.25** | 98.18 | **98.55** |

**Table 5. Results of comparison of current advanced networks on OTU_CEUS.**

| Model | Parameters | mAcc(%) | mIoU(%) | mDice(%) | mPr(%) | mRe(%) |
|---|---|---|---|---|---|---|
| M$^2$SNet [44] | 29 M | 89.08 | 72.18 | 73.50 | 91.85 | 94.50 |
| FCBFormer [29] | 52 M | 90.44 | 73.85 | 74.74 | 94.71 | 93.55 |
| SegFormer [30] | 84 M | 91.31 | 75.40 | 75.57 | **95.66** | 93.71 |
| SSFormer [17] | 66 M | 91.01 | 75.54 | 76.01 | 94.25 | 94.60 |
| PMFFNet | **23 M** | **91.65** | **77.03** | **78.02** | 94.86 | **94.82** |

Similarly, our model demonstrates higher mDice scores, surpassing M$^2$SNet, FCBFormer, Seg-Former, and SSFormer by 4.52%, 3.28%, 2.45%, and 2.01% respectively. These results substantiate that our model not only achieves superior generalization performance across different datasets but also does so with a significantly lower parameter count.

Similarly, the upper part of Fig 8 represents the testing set results obtained from training on the OTU_2D dataset, while the lower part depicts the OTU_CEUS testing results derived from training on the OTU_2D dataset. Observing Fig 7, we can discern that only our model and the M$^2$SNet model achieve reasonably accurate segmentation of the challenging feature. However, our model surpasses M$^2$SNet in terms of segmentation performance. Conversely, the remaining models fail to accurately segment the feature. Notably, M$^2$SNet incorporates a multi-scale subtraction module to enhance its ability to learn complex features. Similarly, our model incorporates MFB module to facilitate the learning of challenging features. Despite these additions, the overall performance of M$^2$SNet is not as satisfactory, as it falls short in com-parison to our model. Our model not only outperforms M$^2$SNet in segmentation accuracy but also achieves this with a lower parameter count.

## Ablation experiments

In order to further demonstrate the effectiveness of our proposed method, we conducted an ablation study to analyze the impact of each module on the performance of our model PMFFNet. We evaluated the effect of each module using metrics such as mAcc, mIoU, mDice, mPrecision, and mRecall. The first ablation experiment, presented in Table 6, involved training and testing on the OTU_2D dataset. The second experiment, shown in Table 7, involved training on the OTU_2D dataset and testing on the OTU_CEUS dataset.

In the ablation study, we first utilize the ViTAEv2+VSA model to obtain feature pyramid and subsequently generates segmentation maps using the deepest features. Next, we

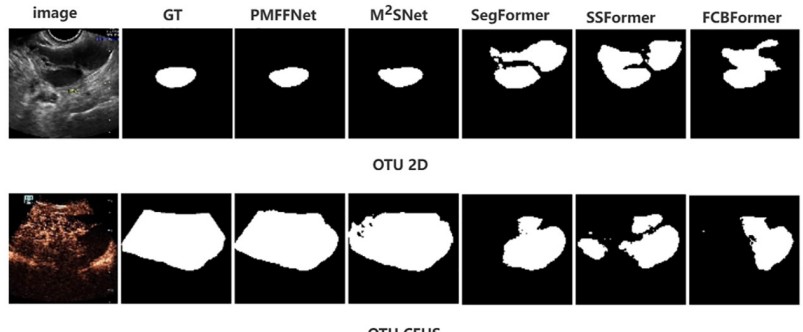

**Fig 8. The segmentation results of advanced networks and our model.**

**Table 6. Results of the ablation study of the proposed model PMFFNet on OTU_2D.**

| Model | mAcc(%) | mIoU(%) | mDice(%) | mPr(%) | mRe(%) |
|---|---|---|---|---|---|
| ViTAEv2+VSA | 96.12 | 87.30 | 82.91 | 97.73 | 97.60 |
| ViTAEv2+VSA +decoder | 96.65 | 90.06 | 86.09 | 97.95 | 98.11 |
| PMFFNet | **97.24** | **91.15** | **87.25** | **98.18** | **98.55** |

**Table 7. Results of the ablation study of the proposed model PMFFNet on OTU_CEUS.**

| Model | mAcc(%) | mIoU(%) | mDice(%) | mPr(%) | mRe(%) |
|---|---|---|---|---|---|
| ViTAEv2+VSA | 89.51 | 69.95 | 68.74 | 92.65 | 91.94 |
| ViTAEv2+VSA+decoder | 90.63 | 74.69 | 73.70 | 94.56 | 94.02 |
| PMFFNet | **91.65** | **77.03** | **78.02** | **94.86** | **94.82** |

incorporated our decoder into V2S, resulting in the ViTAEv2+VSA+decoder model, which further improved network performance. The final model, PMFFNet, by adding our proposed MFB module, we achieved the best results across all metrics. In the first experiment, the inclusion of the decoder led to considerable improvements, with the first three metrics experiencing increases of 0.53%, 3.03%, and 3.18% respectively. This highlights the effectiveness of our inter-layer multi-scale decoder. Fur-theremore, when the MFB module was added, incorporating the ERF and LE modules, the first three metrics increased by 0.59%, 1.09%, and 1.16% respectively. This demonstrates the positive impact of the MFB module on performance.

When we assessed the generalization performance on the OTU_CEUS dataset, significant improvements were observed. According to Table 7, after using the decoder, the mAcc, mIoU, and mDice metrics improved by 1.12%, 4.74%, and 4.96% respectively. Furthermore, the addition of the MFB module resulted in further improvements, with the three metrics increasing by 1.02%, 2.34%, and 4.32% respectively. These re-sults further prove the effectiveness of each module and indicate that our model exhibits strong learning ability within the same dataset and demonstrates robust generalization across different ovarian cancer datasets.

## Conclusion

In this study, we introduce the PMFFNet model, specifically designed for the segmentation of ovarian tumors in ultrasound images. Our proposed network architecture is based on the encoder-decoder architecture, with the encoder employing the VSA integrated into ViTAEv2 model to obtain feature pyramid. Additionally, we pro-pose the MFB module to extract more precise boundary segmentation features. The decoder module fuses and up-samples the MFB module's output iteratively to generate the final segmentation map.

Experimental evaluations conducted on the MMOTU dataset demonstrate that the PMFFNet model exhibits exceptional generalization and learning capabilities. Moreover, our model shows promising potential for advancing other medical image segmentation tasks. Fur-thermore, the heat map analysis of the MFB module validates its effectiveness in suppressing attention dispersion in the Transformer and enhancing the receptive field to capture finer boundary details. The architectural feature pyramid of our model significantly influences the overall segmentation performance. Thus, in future research, we can find a more powerful Transformer backbone network as the encoder of our model to obtain a feature pyramid.

Recently, there has been significant research focused on lightweight medical segmentation networks for point-of-care applications [52, 53]. This is because networks like TransUnet,

which are designed for quality-oriented segmentation, tend to have heavy parameters, computational complexity, and slow inference speed, making them inefficient for fast image segmentation in point-of-care settings. Therefore, it is worth exploring the development of lightweight and high-performance segmentation networks to meet the demands of point-of-care applications. In the future, the second direction is to focus on developing a network that strikes a balance between lightweight design and segmentation performance.

## Author Contributions

**Conceptualization:** Lang Li, Hongbing Ma.

**Data curation:** Hongbing Ma.

**Formal analysis:** Liang He.

**Funding acquisition:** Jing Ma, Gang Sun.

**Investigation:** Jing Ma, Gang Sun.

**Methodology:** Lang Li, Hongbing Ma.

**Project administration:** Liang He, Wenjia Guo.

**Resources:** Liang He.

**Software:** Lang Li.

**Supervision:** Gang Sun.

**Validation:** Wenjia Guo.

**Visualization:** Lang Li, Liang He.

**Writing – original draft:** Lang Li, Hongbing Ma.

**Writing – review & editing:** Lang Li, Hongbing Ma.

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
