## [Decision Letter · Decision Letter 0]

2 Jan 2024

PONE-D-23-34422PMFFNet: Pyramid multi-scale feature fusion network for ovary ultrasound image segmentationPLOS ONE

Dear Dr. Li,

Thank you for submitting your manuscript to PLOS ONE. After careful consideration, we feel that it has merit but does not fully meet PLOS ONE’s publication criteria as it currently stands. Therefore, we invite you to submit a revised version of the manuscript that addresses the points raised during the review process.

We look forward to receiving your revised manuscript.

Kind regards,

Mohamed Hammad, Ph.D.

Academic Editor

PLOS ONE

Journal Requirements:

"Key Research and Development Project of Xinjiang Uygur Autonomous Region (2022B0319-6)"

Reviewers' comments:

Reviewer's Responses to Questions

**Comments to the Author**

1. Is the manuscript technically sound, and do the data support the conclusions?

Reviewer #1: Yes

Reviewer #2: Yes

2. Has the statistical analysis been performed appropriately and rigorously? 

Reviewer #1: Yes

Reviewer #2: No

3. Have the authors made all data underlying the findings in their manuscript fully available?

Reviewer #1: Yes

Reviewer #2: Yes

4. Is the manuscript presented in an intelligible fashion and written in standard English?

Reviewer #1: Yes

Reviewer #2: Yes

5. Review Comments to the Author

Reviewer #1: Dear Authors

The paper titled “PMFFNet: Pyramid multi-scale feature fusion network for ovary ultrasound image segmentation”. The paper addresses important issues however it needs improvements.

1. The title is quite long and complex.

2. The Abstract has a lack of flow and is full of mistakes.

3. There are some limitations in the abstract that need to be resolved:

3.1 Limited Clinical Validation

3.2 Dataset Specificity

3.3 No mention of the model's scalability, resource demands, or processing time, critical for assessing practicality in different computational environments.

3.4 Clinical Impact Assessment:

4. The introduction section is the weakest part of this manuscript.

4.1 Limited Scope Description: The introduction lacks clarity on the scope of clinical validation, leaving the extent of practical application of medical images in diagnosis planning and surgical evaluation ambiguous.

4.2 Population-Specific Focus: The focus on Chinese ovarian cancer cases raises concerns about the generalizability of findings to broader populations, limiting the global applicability of proposed diagnostic and treatment strategies.

4.3 Challenges in Image Segmentation: While acknowledging the challenges of ovarian cancer image segmentation, specific issues faced by the proposed model are not detailed, leaving uncertainties about its performance in diverse clinical scenarios.

4.4 Empirical Judgment vs. AI Transition: The positive shift from expert judgment to artificial intelligence is presented without addressing potential drawbacks, such as interpretability issues or biases in the model resulting from training data.

4.5 Ethical Oversight: Ethical considerations related to patient data privacy, consent, and potential biases in the data are not discussed, raising concerns about the responsible implementation of AI technology in medical contexts.

4.6 Transformer Integration Rationale: Introduction of the pyramid Transformer architecture briefly mentions Varied-Size Window Attention (VSA) without explicit discussion of its advantages or limitations.

4.7 Complex Contributions Summary: The summary of contributions is detailed but may be overly complex, potentially limiting understanding, especially for a broader audience, including those not well-versed in the field.

4.8 Practical Impact Assessment Absence: Despite highlighting contributions, the introduction does not discuss the practical impact of the proposed model on improving diagnostic efficiency, treatment planning, or patient outcomes.

4.9 Limited Limitations Disclosure

4.10 The introduction lacks a section discussing potential limitations of the proposed model, a crucial aspect for understanding constraints and areas for potential improvement.Support from Prior Studies: Some of the citations provided in the introduction lack motivation to use AI/DL, making it challenging for readers to assess the recency and relevance of these studies. The introduction section should include what makes DL so famous across multiple areas, and which type of recent applications it must justify its utilization in the present work. I suggest adding the following as dcnnbt: a novel deep convolution neural network-based brain tumor classification model; brain tumor identification using data augmentation and transfer learning approach.; u-net-based models towards optimal mr brain image segmentation; an intuitionistic approach for the predictability of anti‑angiogenic inhibitors in cancer diagnosis;

4.11 Lack of Research Objectives: The introduction does not explicitly state the research objectives or goals. Readers would benefit from a clearer understanding of what the authors aim to achieve with their proposed method.

4.12 Need for Structure: The introduction could benefit from better structuring to guide readers through the logical flow of ideas, from the problem statament, need of DL and its various applications and then general field of objectives.

5. Some potential drawbacks or limitations of the described methodology:

5.1 Complex Architecture: PMFFNet introduces intricate components like VSA, MFB, and HDC, potentially increasing computational demands.

5.2 Training Challenges: Deep learning models, especially with transformers, require substantial training data and resources, posing challenges in data availability and resource allocation.

5.3 Parameter Tuning Complexity: Components like VSA and MFB add complexity, necessitating meticulous parameter tuning, which may be time-consuming.

5.4 Overfitting Risk: Specialized models like PMFFNet may be susceptible to overfitting on specific datasets, limiting generalization to diverse scenarios.

5.5 Interpretability Concerns: The model's complexity, driven by HDC and VSA, could compromise interpretability, impacting user understanding.

5.6 Resource-Intensive Deployment: Deploying PMFFNet in resource-constrained environments may be challenging due to its resource-intensive nature.

5.7 Pre-training Dependency: Relying on ImageNet pre-training for ViTAEv2 introduces dependencies, with implications for the model's effectiveness.

5.8 Limited Generalization Evaluation: Evaluation of specific datasets raises questions about PMFFNet's generalization across diverse imaging conditions.

5.9 Potential Bias in Training: Lack of diverse training data may lead to biased performance, affecting the model's robustness across demographics.

6. Some potential drawbacks or limitations of the implementation

6.1 Data Limitations and Underfitting Risk: The MMOTU dataset's limited size and susceptibility to underfitting may impact the model's ability to generalize effectively.

6.2 Resolution Standardization Challenges: Resizing input images to 384x384 for consistency raises concerns about the potential loss of information or distortion, especially in images with diverse scales.

6.3 Data Augmentation Dependency: Relying on horizontal and vertical flips for data augmentation may not be sufficient for capturing complex variations in the data, potentially limiting model robustness.

6.4 Loss Function Sensitivity: Solely employing binary cross-entropy (BCE) and Dice loss functions may oversimplify the optimization landscape, possibly affecting the model's adaptability to diverse tasks.

6.5 Dataset Specificity in Training: Training and evaluation on the OTU 2D dataset might limit the model's versatility, raising questions about its performance across broader medical imaging domains.

6.6 Limited Generalization Assessment: Assessing generalization on the OTU CEUS dataset may not provide a comprehensive understanding of the model's adaptability to various imaging modalities or medical conditions.

6.7 Resource-Intensive Training: Training for 100 epochs with a batch size of 8 on an NVIDIA GeForce RTX 3090 GPU may be resource-intensive, restricting accessibility for users without high-end GPUs.

6.8 Optimizer Sensitivity: Depending on the AdamW optimizer with an initial learning rate of 1e-4 may pose challenges in finding an optimal balance between convergence speed and stability.

6.9 Learning Rate Adjustment Strategy: The learning rate adjustment based on consecutive epochs without improvement in mDice may be sensitive to fluctuations, potentially impacting the model's convergence trajectory.

6.10 Parameter Standardization Challenges: While setting parameters uniformly for fair comparison is intended, it may not account for the nuances of specific tasks or datasets, potentially limiting the model's adaptability in diverse scenarios.

6.11 Limited mention of additional metrics and comprehensive model comparisons hinders a thorough assessment of performance.

7. Limitations and the future scope should be added with more clarity.

Reviewer #2: It is an interesting research to apply machine vision to ovary ultrasound image segmentation. In this paper, multi-scale feature fusion block and up-sampling strategy on different inter-layer scales are proposed to enhance the accuracy of segmentation. The experimental results validate the performance of the proposed network.

6. PLOS authors have the option to publish the peer review history of their article (what does this mean?). If published, this will include your full peer review and any attached files.

Reviewer #1: No

Reviewer #2: No

---

## [Author Response · Author response to Decision Letter 0]

25 Jan 2024

Dear editor and reviewers,

Thank you for giving me the opportunity to revise the paper. We have put forth our best efforts to enhance the manuscript and have made changes. These changes do not affect the content and framework of the paper. We sincerely appreciate the dedicated work of the Editors/Reviewers and hope that the corrections will meet with approval. Once again, thank you very much for your comments and suggestions.

Sincerely,

Lang Li

Review Comments to the Author and Response to Reviewers

Reviewer #1: Dear Authors

The paper titled “PMFFNet: Pyramid multi-scale feature fusion network for ovary ultrasound image segmentation”. The paper addresses important issues however it needs improvements.

Answer: 

Thank you for your suggestions. Your comments are highly valuable to our article. Each of these comments has greatly contributed to improving the quality of our work. Following the revision, we have provided a point-by-point response letter to address your suggestions.

1. The title is quite long and complex.

Answer: 

Based on your suggestion, we have revised our title and highlighted it in red, hoping to meet your requirements.

2. The Abstract has a lack of flow and is full of mistakes.

Answer: 

Thank you very much for your suggestions. We have made modifications to our abstract section and highlighted it in yellow based on your feedback, and we hope it meets your requirements.

3. There are some limitations in the abstract that need to be resolved:

3.1 Limited Clinical Validation

Answer: 

You are correct. However, this paper is more focused on the image processing of ovarian tumors, and thus does not delve into clinical discussions. Nevertheless, we have clinical doctors(Wenjia Guo, Gang Sun) from cancer hospitals among our authors, and all the results have been approved by them and are helpful in their clinical practice.

3.2 Dataset Specificity

Answer: 

Due to privacy and other issues, obtaining medical imaging datasets can be challenging. Additionally, medical imaging of datasets ovarian tumors is more challenging to obtain because this type of cancer occurs exclusively in the female population, and there is relatively less research and work focused on it compared to other types of cancer. Currently, we only have access to a dataset from the Chinese region. We acknowledge that we have not conducted experiments on other regions or populations. If we have access to datasets from other regions or populations in the future, we will definitely validate our methods.

3.3 No mention of the model's scalability, resource demands, or processing time, critical for assessing practicality in different computational environments.

Answer: 

Thank you very much for your suggestions. It is indeed critical to consider the model's scalability, resource demands, and processing time when assessing its practicality in different computational environments. However, our main focus is on improving the accuracy of ovarian tumor segmentation, particularly in scenarios where there is a high demand for segmentation performance. In the initial version of the abstract, we briefly mentioned the parameters because in our comparative models, our model has the fewest parameters except for ResUnet++, while achieving the best segmentation results. To avoid misunderstandings, we have removed the discussion about parameters from the abstract. Additionally, we have supplemented information about our model's resource demands and processing time in the experimental section and highlighted it in pink to provide readers with a better understanding of the model's practicality.

3.4 Clinical Impact Assessment:

Answer: 

You are correct. However, this paper is more focused on the image processing of ovarian tumors, and thus does not delve into clinical discussions. Nevertheless, we have clinical doctors(Wenjia Guo, Gang Sun) from cancer hospitals among our authors, and all the results have been approved by them and are helpful in their clinical practice.

4. The introduction section is the weakest part of this manuscript.

Answer: 

Thank you very much for your suggestions. You are absolutely right. Our introduction section did have several weak points. Following your advice, we have made revisions in accordance with your recommendations, hoping to meet your expectations.

4.1 Limited Scope Description: The introduction lacks clarity on the scope of clinical validation, leaving the extent of practical application of medical images in diagnosis planning and surgical evaluation ambiguous.

Answer: 

Following your advice, we have revised the introduction section of the original manuscript and incorporated information about the practical applications of ovarian tumor segmentation, with specific emphasis highlighted in green. We hope these changes meet your expectations.

4.2 Population-Specific Focus: The focus on Chinese ovarian cancer cases raises concerns about the generalizability of findings to broader populations, limiting the global applicability of proposed diagnostic and treatment strategies.

Answer: 

You are right. However, due to privacy and other issues, obtaining medical imaging datasets can be challenging. Additionally, medical imaging datasets of ovarian tumors is more challenging to obtain because this type of cancer occurs exclusively in the female population, and there is relatively less research and work focused on it compared to other types of cancer. Currently, we only have access to a dataset from the Chinese region, which may raise concerns about the applicability of our research to other regions or even globally. While there may be variations in ovarian tumors among women worldwide, there are generally similar characteristics. In the future, we will make efforts to obtain datasets from other regions or different ethnicities, and if available, we will definitely validate our methods using those datasets.

4.3 Challenges in Image Segmentation: While acknowledging the challenges of ovarian cancer image segmentation, specific issues faced by the proposed model are not detailed, leaving uncertainties about its performance in diverse clinical scenarios.

Answer: 

Thank you very much for your suggestions. Our model is primarily designed to improve the performance of ovarian tumor segmentation and assist doctors in diagnosis. Although we acknowledge that there may be uncertainty in different clinical presentations, the segmentation task relies on medical images generated by specialized medical equipment. In various clinical scenarios, medical images generated by the same type of equipment are generally consistent and objective. Therefore, the segmentation performance is expected to be relatively stable.

4.4 Empirical Judgment vs. AI Transition: The positive shift from expert judgment to artificial intelligence is presented without addressing potential drawbacks, such as interpretability issues or biases in the model resulting from training data.

Answer: 

Thank you for your suggestions. Your insights are absolutely correct. The transition from expert judgment to the positive integration of artificial intelligence indeed has some unresolved potential drawbacks, which have been ongoing challenges in the field of AI. However, there is increasing effort being made to address these limitations and shortcomings. For example, addressing the issue of interpretability, Marco et al. [1] proposed a novel interpretability technique called “LIME”. This technique allows for the interpretation of predictions made by any classifier or regression model by learning an interpretable model around the prediction results in an explainable, accurate, and reliable manner. Zeiler [2] proposed visualization techniques that utilize deconvolution to observe the evolution and impact of features during the training process, providing some insights into the internal structure and parameters of CNNs. Zhang Quanshi and Zhu Song-chun [3] provided a comprehensive review of recent research advancements in neural network interpretability.

Indeed, variations in training data can introduce model bias. However, we can only strive to do our best within the available range of data. Additionally, we can mitigate the model bias caused by different training data by leveraging large-scale datasets or multimodal datasets. For example, the recent SA-1B dataset in SAM[4] is a vast dataset that provides researchers with a valuable resource for conducting research and evaluation on segmentation tasks.

4.5 Ethical Oversight: Ethical considerations related to patient data privacy, consent, and potential biases in the data are not discussed, raising concerns about the responsible implementation of AI technology in medical contexts.

Answer:

Indeed, ethical supervision is highly important. However, we are using publicly available datasets, and the authors of the dataset have already discussed the ethical considerations in the original paper. The original text is as follows: "When collecting the images, we find that all images contain the private information of patients. According to the privacy policy, we manually crop the images to remove the private information of patients and make sure that the published MMOTU image dataset will not contain any private information."[5]. Additionally, we have taken your advice into consideration, and we have added a statement regarding ethical considerations in the dataset section in the paper, highlighting it in light blue.

4.6 Transformer Integration Rationale: Introduction of the pyramid Transformer architecture briefly mentions Varied-Size Window Attention (VSA) without explicit discussion of its advantages or limitations.

Answer: 

Thank you very much for your suggestions. Indeed, the VSA was only briefly mentioned in this paper without much discussion. Therefore, we have added relevant content to the Encoder section in order to meet your requirements, highlighting it in orange.

4.7 Complex Contributions Summary: The summary of contributions is detailed but may be overly complex, potentially limiting understanding, especially for a broader audience, including those not well-versed in the field.

Answer: 

Your advice is very reasonable. We have made modifications to the contributions based on your suggestions, highlighting them in light purple, hoping to meet your requirements.

4.8 Practical Impact Assessment Absence: Despite highlighting contributions, the introduction does not discuss the practical impact of the proposed model on improving diagnostic efficiency, treatment planning, or patient outcomes.

Answer: 

You are correct. However, this paper primarily focuses on the image processing of ovarian tumors. The aim is to improve the segmentation accuracy of ovarian cancer tumors, as higher segmentation accuracy can better assist clinicians in diagnosing the condition and making treatment plans accordingly. And we have clinical doctors(Wenjia Guo, Gang Sun) from cancer hospitals among our authors, and all the results have been approved by them and are helpful in their clinical practice. Additionally, we have also made modifications to the content in the introduction section, hoping to meet your requirements.

4.9 Limited Limitations Disclosure

Answer:

For our medical image segmentation task, the required data is already publicly available, so there may not be any relevant disclosure restrictions. Additionally, if you need the code for our model, we would be happy to provide it to you. Lastly, if you require any further disclosures, we will make our best efforts to provide them to you.

4.10 The introduction lacks a section discussing potential limitations of the proposed model, a crucial aspect for understanding constraints and areas for potential improvement.Support from Prior Studies: Some of the citations provided in the introduction lack motivation to use AI/DL, making it challenging for readers to assess the recency and relevance of these studies. The introduction section should include what makes DL so famous across multiple areas, and which type of recent applications it must justify its utilization in the present work. I suggest adding the following as dcnnbt: a novel deep convolution neural network-based brain tumor classification model; brain tumor identification using data augmentation and transfer learning approach.; u-net-based models towards optimal mr brain image segmentation; an intuitionistic approach for the predictability of anti angiogenic inhibitors in cancer diagnosis;

Answer: 

Thank you very much for your suggestions. Firstly, we have included the potential limitations of the model in the conclusion section and further clarified future work, highlighting it in gray. Secondly, we have added all the relevant papers regarding the motivation for using AI or DL that you mentioned to our introduction section, highlighting them in light orange.

4.11 Lack of Research Objectives: The introduction does not explicitly state the research objectives or goals. Readers would benefit from a clearer understanding of what the authors aim to achieve with their proposed method.

Answer:

Your advice is very reasonable. In order to provide greater clarity to the readers, we have supplemented and clarified the research objectives in the introduction section, highlighting them in light red.

4.12 Need for Structure: The introduction could benefit from better structuring to guide readers through the logical flow of ideas, from the problem statament, need of DL and its various applications and then general field of objectives.

Answer:

Thank you for your suggestions. We have made modifications to the structure of the introduction section according to your advice. Additionally, in order to improve the logical flow, we have added some supplementary content, highlighted in blue-black.We sincerely appreciate your valuable suggestions regarding the structure of the introduction, and we have greatly benefited from them.

5. Some potential drawbacks or limitations of the described methodology:

5.1 Complex Architecture: PMFFNet introduces intricate components like VSA, MFB, and HDC, potentially increasing computational demands.

Answer:

You are right, some of the components we added do potentially increase computational demands, but they have resulted in improved segmentation accuracy. Our main objective is to enhance the quality of ovarian tumor segmentation, which is crucial for clinical doctors to assess patients' conditions. In future work, we will also consider developing a lightweight network that maintains high segmentation performance.

5.2 Training Challenges: Deep learning models, especially with transformers, require substantial training data and resources, posing challenges in data availability and resource allocation.

Answer: 

You are right, Transformer models do require a significant amount of data resources. However, their pre-trained models on ImageNet have learned extensive visual prior knowledge, and they exhibit remarkable generality in downstream tasks. Additionally, the ViTAEv2[6] paper's "Data efficiency and training efficiency" section also demonstrates the effectiveness of ViTAE in learning with limited training data and fewer training epochs. Of course, using more data is always beneficial. Furthermore, data augmentation is also an effective means to alleviate data requirements.

5.3 Parameter Tuning Complexity: Components like VSA and MFB add complexity, necessitating meticulous parameter tuning, which may be time-consuming.

Answer:

Indeed, components such as VSA and MFB do add complexity, and parameter tuning can be time-consuming. However, these components have relatively few tunable parameters.

1、Based on the ablation experiments conducted by the original authors of VSA[7], it can be observed that the performance of using VSA at each stage is optimal. Therefore, we employ VSA at each stage. Additionally, the window size of VSA is learned from the data and does not require manual parameter setting from us.

2、The dilation parameters of the MFB module need to satisfy the HDC principle. In this paper, we chose a kernel size of 3, and there are only a few dilation parameters that satisfy the HDC principle.

For K=3 and the dilation parameters r=[1,2,9], when we substitute them into the formula, we get M2 = 5, which is indeed greater than 3. Therefore, in this case, the dilation parameters do not satisfy the HDC principle.While it is possible to try other kernel sizes, in general, a kernel size of 3 is a commonly used and suitable choice.

5.4 Overfitting Risk: Specialized models like PMFFNet may be susceptible to overfitting on specific datasets, limiting generalization to diverse scenarios.

Answer:

You are correct, overfitting is also a very common issue.In addressing the overfitting issue, we have implemented data augmentation techniques, which can alleviate the problem to some extent. Additionally, in ViTAEv2, we have also utilized dropout to reduce the network's reliance on individual neurons, thereby enhancing its generalization ability. Additionally, in the future, if we can obtain sufficient data, we plan to construct a large multimodal dataset to improve the model's generalization ability across different scenarios.

5.5 Interpretability Concerns: The model's complexity, driven by HDC and VSA, could compromise interpretability, impacting user understanding.

Answer:

Indeed, that is true. Regarding the VSA component, we have added some additional content in the Encoder section to provide a better understanding for users, highlighting it in orange. Additionally, HDC is a principle for selecting dilation parameters, and in our model, it is represented solely through the parameters. By using the formula, we can obtain appropriate dilation parameters, and we have also provided an explanation of the HDC principle in the paper.

5.6 Resource-Intensive Deployment: Deploying PMFFNet in resource-constrained environments may be challenging due to its resource-intensive nature.

Answer:

Indeed, PMFFNet may present challenges in resource-constrained environments. However, our main objective is to improve the segmentation accuracy of ovarian cancer to better assist clinicians in improving diagnostic efficiency. In future work, we also plan to develop a lightweight and high-performance network specifically designed for resource-limited environments.

5.7 Pre-training Dependency: Relying on ImageNet pre-training for ViTAEv2 introduces dependencies, with implications for the model's effectiveness.

Answer:

You are correct. However, using pre-trained models is very common in deep learning, and it is often seen in various visual tasks such as Swin Transformer[8], ResNet[9], etc., where the models are pre-trained on ImageNet and then tested as backbones for downstream tasks. The results have shown that using these pre-trained backbones for other visual tasks is highly effective. Additionally, in our comparative experiments, the original papers of models such as SSFormer[10] and FCBFormer[11] also used pre-trained models. Therefore, for consistency, we also need to use pre-trained models.

5.8 Limited Generalization Evaluation: Evaluation of specific datasets raises questions about PMFFNet's generalization across diverse imaging conditions.

Answer:

You make a valid point, but our method primarily focuses on ultrasound images. If we have datasets in the future that include different imaging conditions, we will validate its generalization performance. Additionally, addressing generalization issues across various imaging modalities can be challenging. It may require a large-scale multimodal dataset, which can be difficult to construct due to privacy concerns and other factors in medical imaging. In the future, if there are sufficient datasets available, we will also attempt to construct the multimodal dataset for everyone's use.

5.9 Potential Bias in Training: Lack of diverse training data may lead to biased performance, affecting the model's robustness across demographics.

Answer:

Thank you very much for your suggestions. You are right. However, due to privacy and other issues, obtaining medical imaging datasets can be challenging. Additionally, medical imaging datasets of ovarian tumors is more challenging to obtain because this type of cancer occurs exclusively in the female population, and there is relatively less research and work focused on it compared to other types of cancer. Currently, we only have a dataset from the Chinese region, which may introduce potential biases in terms of other regions or ethnicities. While there may be variations in ovarian tumors among women worldwide, there are generally similar characteristics. And in the future, We will make every effort to acquire datasets from other regions or populations, and if available, we will validate our method on those datasets.

6. Some potential drawbacks or limitations of the implementation

6.1 Data Limitations and Underfitting Risk: The MMOTU dataset's limited size and susceptibility to underfitting may impact the model's ability to generalize effectively.

Answer:

Thank you very much for your suggestions. Due to the difficulty in obtaining medical imaging datasets, we acknowledge the limitations of the currently available dataset. We have taken into account the limited data by employing data augmentation techniques, and our model is not a simple model, which allows it to better learn features and minimize the risk of underfitting. If we have access to more datasets in the future, we will validate our methods.

6.2 Resolution Standardization Challenges: Resizing input images to 384x384 for consistency raises concerns about the potential loss of information or distortion, especially in images with diverse scales.

Answer:

You are correct that modifying the original image size can potentially result in information loss. However, in our dataset, the lesion regions are not small targets but rather large areas, so the distortion is limited. Additionally, our ultrasound images have varying dimensions, with widths ranging from 302 to 1135 pixels and heights ranging from 226 to 794 pixels. Using larger input sizes would significantly increase computational requirements. Therefore, we have adopted the size of 384×384, which is already a commonly used large input size in visual tasks. Furthermore, in the original paper of the MMOTU dataset, the input images were also resized and cropped to 384×384. Moreover, training is typically done using batches rather than individual images, and having the same size within a batch facilitates computational efficiency.

6.3 Data Augmentation Dependency: Relying on horizontal and vertical flips for data augmentation may not be sufficient for capturing complex variations in the data, potentially limiting model robustness.

Answer:

You are right. However, the use of data augmentation is indeed a commonly employed practice in the field of computer vision. It not only enhances the model's generalization ability but also helps prevent overfitting. In addition to incorporating flips, we have also utilized methods such as affine transformations. 

6.4 Loss Function Sensitivity: Solely employing binary cross-entropy (BCE) and Dice loss functions may oversimplify the optimization landscape, possibly affecting the model's adaptability to diverse tasks.

Answer:

Thank you very much for your suggestions. Indeed, using a simple loss function can simplify the optimization environment. However, the combination of BCELoss and Dice loss is commonly used in medical segmentation tasks. For example, both SSFormer and FCBFormer also use the BCE+Dice loss function. Additionally, excessively complex loss functions can make model training more challenging.

6.5 Dataset Specificity in Training: Training and evaluation on the OTU 2D dataset might limit the model's versatility, raising questions about its performance across broader medical imaging domains.

Answer:

You are right. However due to privacy and other issues, obtaining medical imaging datasets can be challenging. Additionally, medical imaging datasets of ovarian tumors is more challenging to obtain because this type of cancer occurs exclusively in the female population, and there is relatively less research and work focused on it compared to other types of cancer. Currently, we only have a dataset from the Chinese region, which may introduce potential biases in terms of other regions. While there may be variations in ovarian tumors among women worldwide, there are generally similar characteristics. We will make every effort to acquire datasets from other regions in the future, and if available, we will validate our method on those datasets.

6.6 Limited Generalization Assessment: Assessing generalization on the OTU CEUS dataset may not provide a comprehensive understanding of the model's adaptability to various imaging modalities or medical conditions.

Answer:

Thank you very much for your suggestions. You are right. However our method primarily focuses on ultrasound images. If we have datasets in the future that include different imaging conditions, we will validate its generalization performance. Additionally, addressing generalization issues across various imaging modalities can be challenging. It may require a large-scale multimodal dataset, which can be difficult to construct due to privacy concerns and other factors in medical imaging. In the future, if there are sufficient datasets available, we will also attempt to construct the multimodal dataset for everyone's use.

6.7 Resource-Intensive Training: Training for 100 epochs with a batch size of 8 on an NVIDIA GeForce RTX 3090 GPU may be resource-intensive, restricting accessibility for users without high-end GPUs.

Answer:

Indeed, high-end GPUs like the 3090 are not universally available. We trained on the 3090 primarily to expedite the training process. However, if the resources are limited, it is not necessary to choose the 3090 specifically. As long as the GPU's memory is sufficient, training can still be performed, albeit at a slower pace compared to the 3090.

6.8 Optimizer Sensitivity: Depending on the AdamW optimizer with an initial learning rate of 1e-4 may pose challenges in finding an optimal balance between convergence speed and stability.

Answer:

Thank you very much for your suggestions. The initial learning rate of 1e-4 with the AdamW optimizer is indeed a common setting and widely used in many computer vision tasks. For example, in works like SSFormer and FCBFormer, the initial learning rate is set to 1e-4 as well. Additionally, through experimentation, we have observed that using a learning rate of 1e-4 leads to fast convergence within the first 10 epochs, and then it gradually converges more slowly and reaches a stable state.

6.9 Learning Rate Adjustment Strategy: The learning rate adjustment based on consecutive epochs without improvement in mDice may be sensitive to fluctuations, potentially impacting the model's convergence trajectory.

Answer:

Our approach is to modify the learning rate only if there is no improvement in the mDice score within 10 epochs, and if there is a single instance of improvement, the counting starts over. Through experimentation, we have found that this strategy allows for more thorough learning at a certain learning rate, and the mDice score continues to improve as the learning rate decreases. Additionally, this strategy is also commonly employed, as seen in the case of FCBFormer, for example.

6.10 Parameter Standardization Challenges: While setting parameters uniformly for fair comparison is intended, it may not account for the nuances of specific tasks or datasets, potentially limiting the model's adaptability in diverse scenarios.

Answer:

Thank you very much for your suggestions, and you are right. However, in our experiments, we kept the loss function, image input size, and other common settings consistent, and we did not make any modifications to the model architecture.

6.11 Limited mention of additional metrics and comprehensive model comparisons hinders a thorough assessment of performance.

Answer:

In segmentation tasks, the most important evaluation metrics are ACC, Dice, and IoU, all of which I have used in this study. Additionally, these metrics are also employed in FCBFormer.

7. Limitations and the future scope should be added with more clarity.

Answer:

Thank you very much for your suggestions. In order to make the article clearer, we have made additions to the conclusion section based on your advice, highlighting them in gray.

Thank you once again for your suggestions on this paper. Your feedback is of great significance to our article, and we hope that the revised version meets your requirements. Have a wonderful day!

Reviewer #2: It is an interesting research to apply machine vision to ovary ultrasound image segmentation. In this paper, multi-scale feature fusion block and up-sampling strategy on different inter-layer scales are proposed to enhance the accuracy of segmentation. The experimental results validate the performance of the proposed network.

Answer:

Thank you very much for your encouragement. Ovarian ultrasound image segmentation is indeed an important area of research, given the high mortality rate associated with ovarian tumors. If we can effectively screen ovarian tumors at an early stage, it can significantly reduce the mortality rate of ovarian cancer. Portable devices such as POCUS (Point-of-Care Ultrasound)[12] provide a convenient way to acquire ultrasound images and perform segmentation directly. However, these point-of-care devices require lightweight models. Therefore, in our future work, we plan to develop a more lightweight and high-performance network to meet the demands of point-of-care scenarios. Once again, thank you for your encouragement, and we will strive to excel in our segmentation task. Have a wonderful day!

References

[1] Ribeiro, M. T., Singh, S., Guestrin, C. " Why should i trust you?" Explaining the predictions of any classifier. In Proceedings of the 22nd ACM SIGKDD international conference on knowledge discovery and data mining, 2016, August;pp. 1135-1144.

[2] Zeiler, M. D., Fergus, R. Visualizing and understanding convolutional networks. In Computer Vision–ECCV 2014: 13th European Conference, Zurich, Switzerland, September 6-12, 2014;pp. 818-833. 

[3] Zhang, Quan-shi, and Song-Chun Zhu. Visual interpretability for deep learning: a survey. Frontiers of Information Technology & Electronic Engineering 19.1 (2018): 27-39.

[4] Kirillov A, Mintun E, Ravi N, et al. Segment anything. arXiv 2023;arXiv:2304.02643,.

[5] Zhao Q, Lyu S, Bai W, Cai L, Liu B, Wu M, et al. A Multi-Modality Ovarian Tumor Ultrasound Image Dataset for Unsupervised Cross-Domain Semantic Segmentation. arXiv preprint arXiv:220706799. 2022;.

[6] Zhang Q, Xu Y, Zhang J, Tao D. Vitaev2: Vision transformer advanced by exploring inductive bias for image recognition and beyond. International Journal of Computer Vision. 2023; p. 1–22.

[7] Zhang Q, Xu Y, Zhang J, Tao D. Vsa: Learning varied-size window attention in vision transformers. In: European conference on computer vision. Springer; 2022. p. 466–483.

[8] Liu Z, Lin Y, Cao Y, Hu H, Wei Y, Zhang Z, et al. Swin transformer: Hierarchical vision transformer using shifted windows. In: Proceedings of the IEEE/CVF international conference on computer vision; 2021. p. 10012–10022.

[9] He K, Zhang X, Ren S, Sun J. Deep residual learning for image recognition. In: Proceedings of the IEEE conference on computer vision and pattern recognition; 2016. p. 770–778.

[10] Wang J, Huang Q, Tang F, Meng J, Su J, Song S. Stepwise feature fusion: Local guides global. In: International Conference on Medical Image Computing and Computer-Assisted Intervention. Springer; 2022. p. 110–120.

[11] Sanderson E, Matuszewski BJ. FCN-transformer feature fusion for polyp segmentation. In: Annual Conference on Medical Image Understanding and Analysis. Springer; 2022. p. 892–907.

[12]butterflynetwork. Available online: https://www.butterflynetwork.com/iq-ultrasound-individuals (accessed on 10 January 474 2024).

---

## [Decision Letter · Decision Letter 1]

9 Feb 2024

PMFFNet: A hybrid network based on feature pyramid for ovarian tumor segmentation

PONE-D-23-34422R1

Dear Dr. Li,

We’re pleased to inform you that your manuscript has been judged scientifically suitable for publication and will be formally accepted for publication once it meets all outstanding technical requirements.

Kind regards,

Mohamed Hammad, Ph.D.

Academic Editor

PLOS ONE

Additional Editor Comments (optional):

Reviewers' comments:

Reviewer's Responses to Questions

**Comments to the Author**

1. If the authors have adequately addressed your comments raised in a previous round of review and you feel that this manuscript is now acceptable for publication, you may indicate that here to bypass the “Comments to the Author” section, enter your conflict of interest statement in the “Confidential to Editor” section, and submit your "Accept" recommendation.

Reviewer #1: All comments have been addressed

2. Is the manuscript technically sound, and do the data support the conclusions?

Reviewer #1: Yes

3. Has the statistical analysis been performed appropriately and rigorously? 

Reviewer #1: Yes

4. Have the authors made all data underlying the findings in their manuscript fully available?

Reviewer #1: Yes

5. Is the manuscript presented in an intelligible fashion and written in standard English?

Reviewer #1: Yes

6. Review Comments to the Author

Reviewer #1: Dear Authors

I have now completed the review of the revised manuscript. I have observed that the authors put in good efforts to address all the comments satisfactorily.

7. PLOS authors have the option to publish the peer review history of their article (what does this mean?). If published, this will include your full peer review and any attached files.

Reviewer #1: No

---

## [Editor Report · Acceptance letter]

21 Mar 2024

PONE-D-23-34422R1 

PLOS ONE

Dear Dr. Li, 

I'm pleased to inform you that your manuscript has been deemed suitable for publication in PLOS ONE. Congratulations! Your manuscript is now being handed over to our production team.

Kind regards, 

on behalf of

Dr. Mohamed Hammad 

Academic Editor

PLOS ONE